# Early and delayed STAT1-dependent responses drive local trained immunity of macrophages in the spleen

Aryeh Solomon[1], Noa Bossel Ben-Moshe[1], Dotan Hoffman[1], Sébastien Trzebanski[1], Dror Yehezkel[1], Leia Vainman[1], Mihai G Netea[2,3], Roi Avraham[1]*

[1]Department of Immunology and Regenerative Biology, Weizmann Institute of Science, Rehovot, Israel; [2]Department of Internal Medicine and Radboud Center for Infectious Diseases, Radboud University Medical Center, Nijmegen, Netherlands; [3]Department of Immunology and Metabolism, Life and Medical Sciences Institute, University of Bonn, Bonn, Germany

## eLife Assessment

This **important** work advances our understanding of the contribution of tissue-resident immune cells to trained immunity phenotypes. The evidence supporting the claims is **convincing**, with results that will be of interest to immunologists and scientists studying the host-pathogen interface.

*For correspondence:
roi.avraham@weizmann.ac.il

Competing interest: The authors declare that no competing interests exist.

**Abstract** Trained immunity (TI) is the process wherein innate immune cells gain functional memory upon exposure to specific ligands or pathogens, leading to augmented inflammatory responses and pathogen clearance upon secondary exposure. While the differentiation of hematopoietic stem cells (HSCs) and reprogramming of bone marrow (BM) progenitors are well-established mechanisms underpinning durable TI protection, remodeling of the cellular architecture within the tissue during TI remains underexplored. Here, we study the effects of peritoneal Bacillus Calmette–Guérin (BCG) administration to find TI-mediated protection in the spleen against a subsequent heterologous infection by the Gram-negative pathogen *Salmonella* Typhimurium (*S*.Tm). Utilizing single cell RNA-sequencing and flow cytometry, we discerned STAT1-regulated genes in TI-associated resident and recruited splenic myeloid populations. The temporal dynamics of TI were further elucidated, revealing both early and delayed myeloid subsets with time-dependent, cell-type-specific STAT1 signatures. Using lineage tracing, we find that tissue-resident red pulp macrophages (RPM), initially depleted by BCG exposure, are restored from both tissue-trained, self-renewing macrophages and from bone marrow-derived progenitors, fostering long lasting local defense. Early inhibition of STAT1 activation, using specific JAK-STAT inhibitors, reduces both RPM loss and recruitment of trained monocytes. Our study suggests a temporal window soon after BCG vaccination, in which STAT1-dependent activation of long-lived resident cells in the tissue mediates localized protection.

## Introduction

Trained immunity (TI) is defined as the capacity of innate immune cells to recall and modulate their subsequent responsiveness following prior exposure to a diverse array of stimuli (*Arts et al., 2015*). Unlike the adaptive immune system, these changes are antigen-agnostic and are more reflective of sustained alterations in cellular and systemic states. TI has been demonstrated to directly reprogram the metabolic, transcriptional, and epigenetic state of monocytes and macrophages, generating

heightened inflammatory capabilities in-vitro (*Arts et al., 2015*; *Saeed et al., 2014*). As the transient lifespans of these cells are weeks only, this alone is insufficient to explain the sustained protection generated by TI that can last from months to years. To address this limitation, current research has been focused on long-lived multipotent progenitor immune cells.

BCG vaccine (Bacillus Calmette-Guérin), an attenuated strain of *Mycobacterium bovis*, is a potent inducer of TI, which facilitate nonspecific protection against heterologous pathogens in mice, primates, and humans (*Darrah et al., 2020*; *Moorlag et al., 2019*). The effects of intravenous (i.v.) BCG vaccination, where bacilli persist within the bone marrow (BM), drive the expansion of LSK +hematopoietic stem cells (HSCs) and multipotent progenitors (MPPs; *Moorlag et al., 2019*). In conjunction, MPPs undergo biased differentiation towards myelopoiesis, increasing the proliferation of myeloid cells. These effects are driven primarily by interferon activation, particularly that of interferon-gamma (IFNγ) and its downstream transcription factors, STAT1 and IRF1, in MPPs and short-term HSCs, resulting in a transcriptional program of cell cycling (Cdk1, Cdk4, Ccnf) and histone modifications. Furthermore, these newly differentiated circulating cells have enhanced microbicidal properties (*Hasso-Agopsowicz et al., 2024*; *Qadri et al., 2024*). Upon infection, in order to elicit protection against invading agents, trained progenitor immune cells from the BM interact within local immune cell compartments in the tissue. Understanding these mechanisms is particularly relevant given emerging challenges in infectious disease control. The rise of antimicrobial-resistant bacteria demand the development of attenuated strains-specific vaccines (*Hasso-Agopsowicz et al., 2024*), such as the typhoid conjugate vaccine (TCV) for the gram negative *Salmonella* Typhi (*Qadri et al., 2024*). The premise of BCG and trained immunity is providing global heterologous protection against a range of pathogens. However, the broader potential of TI-mediated heterologous protection requires understanding of TI-mediated protection as it translates within the tissue and the local cellular architectures is not completely understood.

Most tissues harbor long-lived resident macrophages capable of self-renewal, independent of BM-derived precursors. These resident populations, including pulmonary alveolar macrophages, Kupffer cells in the liver and red pulp macrophages in the spleen, originate from embryonic erythro-myeloid progenitors (*Kurotaki et al., 2015*). Antimicrobial activity of resident populations control early stages of invasion by pathogens but are than depleted, and the niche is repopulated either by self-renewal (*Hoeffel et al., 2015*) or by progenitors derived from the BM to maintain local functions (*Lai et al., 2018*). Intriguingly, resident alveolar macrophages (AM) have been shown to undergo in-situ training, maintaining an altered phenotype over an extended period (*Zhang et al., 2016*; *Barth et al., 1995*; *Keren-Shaul et al., 2019*). One mechanism driving this training in-situ in AMs involves CX3CR1⁺ T cells, which have been implicated in local reprogramming via IFNγ release. This process has been demonstrated to occur around three weeks after initial BCG vaccination, after which interferon-driven responses occur across the organism, including lung and BM, resulting in viral and mycobacterial protection. It remains unclear however, how during TI, a bridge is formed between peripheral, resident cells and BM-derived recruitment of trained myeloid cells.

In this study, we set out to investigate the interplay of circulating and tissue-level immune cells that mediate protection and remodeling during TI. To induce tissue specific TI, we used a model of intraperitoneal (i.p.) injection of BCG, which rapidly delivers the bacteria to target lymphatic organs (*e.g.,* spleen), and measured protection against subsequent *Salmonella* Typhimurium (*S*.Tm) challenge. Within the spleen's unique structure and cellular composition, we characterized STAT1-mediated, cell type-specific TI signatures. We demonstrate that an initial depletion of resident red-pulp macrophages is followed by repopulation by recruited trained monocytes and a local self-renewing population, contributing to the maintenance of STAT1 signatures and long-lasting protection within the tissue.

## Results

### Intraperitoneal BCG results in heterologous *S*.Tm protection and a distinct myeloid subsets with signatures driven by STAT1

To establish an in-vivo BCG training model targeted to the splenic tissue, and assess the extent of cross pathogen tissue protection conferred, we administered BCG-Pasteur (*Zhang et al., 2016*) (5×10⁶ colony forming units (CFU)) or PBS as a control via i.p injection to 8-week-old female C57BL/6 J mice. Following a 2-week training period, mice were challenged with *S*.Tm (5x10⁵ CFU) through i.p

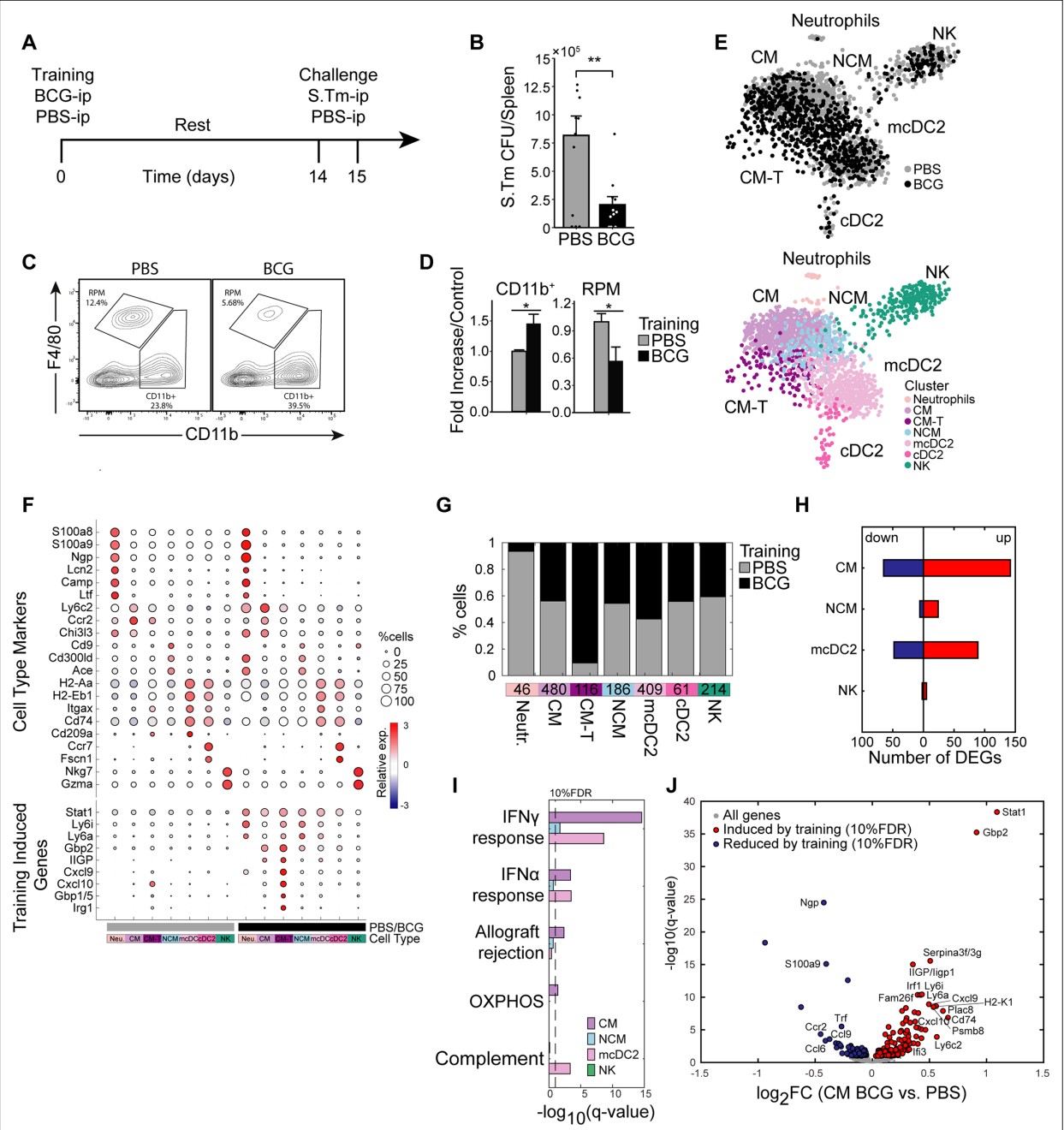

**Figure 1.** Intraperitoneal BCG results in heterologous *S*.Tm protection and a distinct myeloid subsets with signatures driven by STAT1. (**A**) Mouse model of BCG vaccination and S.Tm challenge. (**B**) Splenic S.*Tm* CFU 24 hr post infection between control (n=11) and trained mice (n=11). (**C–D**) Flow cytometry plots of myeloid populations two weeks post vaccination (**C**) and mean percentage fold change of BCG over control for each given gated population percent from the Lin⁻ population (**D**). (**E**) K-nearest neighbors (KNN) plot for total CD11b⁺ single cells sorted from control and BCG mice. Color is based on conditions, or cluster identity. (**F**) Cell markers and training induced genes for each subset. Size and color intensity indicates percentage of cells within a given cluster expressing the gene and average expression. (**G**) Proportions of monocyte subsets based on classifications in (**E**). (**H–I**) Number of DEGs in each cell subset (**H**), and their corresponding gene set enrichment analysis (**I**). (**J**) Volcano plot of DEGs in CM subset. Data in bar graphs are presented as mean ± SEM, with each individual point in (**B**) a biological repeat. Two-tailed *t*-test used for data in (**B**) and (**D**) (*p<0.05, **p<0.01).

The online version of this article includes the following figure supplement(s) for figure 1:

**Figure supplement 1.** Single cell gating strategy and populations-specific DEGs.

inoculation (*Figure 1A*). Mice were sacrificed after 24 hr, and spleen homogenates were cultured on LB agar medium to quantify *S*.Tm load using CFU (*Figure 1B*). Relative to the control group, mice that received BCG exhibited enhanced protection against *S*.Tm, with a four- to fivefold decrease in CFU, indicating TI-mediated splenic protection.

For identification of transcriptional changes in splenic myeloid populations related to TI, we isolated splenocytes and performed staining with CD11b, Ly6C, and F4/80 gating for myeloid mono-nuclear phagocytes (MPs), including classical monocytes (CM; CD11b⁺Ly6C⁺), and CD11b⁺ Ly6C⁻ MPs comprising non-classical monocytes (NCM) and conventional dendritic cells (cDC), and also resident red pulp macrophages (RPM) (CD11b⁻ F4/80⁺; *Figure 1C*, *Figure 1—figure supplement 1A*). We observed an overall expansion of the myeloid compartment due to training, with an increase in the CD11b⁺ subset, and a pronounced reduction of resident RPM (*Figure 1D*). This observed loss of RPM is common during infection and inflammation and has been previously described as the resident macrophage disappearance reaction (*Barth et al., 1995*).

We next determined transcriptional alterations caused by TI across the myeloid compartment. CD11b⁺ cells from trained and naïve mice were isolated by sorting and processed for single-cell RNA-sequencing (scRNA-seq; *Keren-Shaul et al., 2019*). K-nearest neighbor (K-NN) clustering differenti-ated six distinct cell types, designated as CM, NCM, immature DC2 (cDC2), mature DC2 (mcDC2), neutrophils, and NK cells (*Figure 1E*). The identity of each cell type cluster was based on estab-lished transcriptional markers (*Figure 1F*). Additional analysis was performed to identify differentially expressed genes (DEG) unique to training (*Figure 1G–I*; *Supplementary file 1a*). While most popu-lations derived from control or trained mice clustered together by cell type, we identified a subset of trained CM (CM-T) particular to BCG (*Figure 1G*). This subset represents an enhanced state of the STAT1-regulated IFNγ response that characterizes trained CM (e.g. *Gbp2, Ly6a, Cxcl9*, and *Irg1*; *Figure 1J*). Notably, these STAT1 activated genes are known to drive maturation and activation of monocytes and macrophages, enhancing their antimicrobial response (*Hu and Ivashkiv, 2009*).

*Stat1* and its downstream targets (e.g. *Gbp2, Irf1*, and *Ly6a*), exhibited a pronounced elevation also in other myeloid cells (*Figure 1—figure supplement 1B*), aligning with an earlier report that BCG induces BM and HSC remodeling via interferon signaling, resulting in STAT1 upregulation in progeni-tors (*Kaufmann et al., 2018*). However, not all populations were similarly activated in conjunction with STAT1-regulated genes, with gene set enrichment analysis (GSEA) demonstrating differential pathway upregulation across cell types (*Figure 1I*). CM and mcDC2 primarily activated the IFNγ response, while complement associated genes were enriched in mcDC2. NK cells alone showed minimal activa-tion due to training across all detected pathways (*Figure 1—figure supplement 1B*).

We also observed increased SCA-1 (*Ly6a/e*) expression across the monocyte subsets and mcDC2, most notably in NCM (*Figure 1F*; *Figure 1—figure supplement 1B*). SCA-1 (stem cell antigen-1) is a glycosylphosphatidylinositol-anchored cell surface protein commonly used as a marker for murine hematopoietic stem and progenitor cells within the BM. The STAT1 signaling pathway has been shown to be involved in the regulation of SCA-1 expression (*Yang et al., 2017*), suggesting a role for STAT1 in the modulation of SCA-1 during development of TI. Recent studies have also demonstrated a link between SCA-1 and inflammation, with its expression found to be upregulated on a specific subset of Ly6C⁺ monocytes during infection (*Biram et al., 2022*). These SCA-1⁺ monocytes exhibited a pro-inflammatory phenotype, characterized by the production of inflammatory cytokines and chemokines, and were implicated in the amplification of these responses. For Ly6C⁻ NCM, the role of SCA-1 has not yet been studied.

In addition to the upregulation of inflammatory and antimicrobial genes, we also observed a signif-icant downregulation of specific genes in CM-T as a result of training, including *Ccr2*, *S100A8/9*, and *Ngp* (*Figure 1J*). CCR2, a chemokine receptor crucial for monocyte recruitment, was found to be downregulated in response to IFNγ. This reduction in CCR2 expression is mediated by IFNγ-induced mRNA instability, potentially serving to retain monocytes at the site of recruitment and dampen a positive feedback loop (*Penton-Rol et al., 1998*). Similarly, BCG has been shown in other contexts to reduce the expression of S100A8 and S100A9, two calcium-binding proteins that can heterodimerize and stimulate IFNγ production in CD4 +T cells via an IL-10dependent mechanism (*Wang et al., 2023*). By suppressing their activity, runaway signaling and exhaustion is avoided. Lastly, NGP (neutrophilic granule protein), has been implicated in the regulation of inflammation through its ability to block NF-κB signaling (*Liu et al., 2020*). The downregulation of these genes suggests that trained immunity

not only enhances pro-inflammatory responses but also modulates the expression of key regulators to maintain a balanced immune response and prevent excessive inflammation.

## Dynamics of TI-associated subsets and signatures indicates early and delayed kinetics

To gain insight into dynamic processes underlying splenic cell type-specific TI, we conducted a two-month experiment, sacrificing mice at days 3, 14, 30, 45, and 60 post-vaccination with BCG, and challenging with S.Tm at 14 and 60 days post training (*Figure 2A*). We assessed BCG growth in BM and spleen, resistance to *S*.Tm infection, flow cytometry analysis with cell-type-specific training markers (CXCL9 and SCA-1), and bulk RNA-seq. Notably, protection against *S*.Tm persisted for 2 months post-vaccination, albeit with waning resistance over time (*Figure 2B*). While intravenous (i.v) BCG delivery and subsequent localization of the bacterium to the BM were previously noted as crucial for robust training (*Kaufmann et al., 2018*; *Khan et al., 2020*), our results indicate that during i.p administration this process is dispensable, as no BCG were detected in the BM (*Figure 2C*). Conversely, in the spleen, we isolated BCG at all time points, with CFU declining sharply by day 14, reaching the limit of detection by day 30 with minimal bacterium remaining (*Figure 2C*).

Flow cytometry analysis revealed a dynamic process during trained immunity (*Figure 2—figure supplement 1A*). Compared to mean control values, we observed rapid recruitment of CMs in the spleen, starting already at day three after BCG, peaking at day 14 before returning to baseline (*Figure 2D*, *Figure 2—figure supplement 1B*). We also stained cells for CXCL9 expression, as a marker gene of STAT1-mediated TI that defines the CM-T subset, and revealed that this population reaches peak levels at day 14 (*Figure 2—figure supplement 1C*). In a separate experiment to assess early kinetics of CM-Ts, we measured a significant increase of this subset already at five days post BCG (*Figure 2—figure supplement 1D*), indicating an early recruitment of these cells due to the initial response to BCG. NCMs exhibit a different TI signature and kinetic pattern. While their ratio initially declines, they return to steady-state levels by day 30 (*Figure 2D*). NCMs express SCA-1 (*Ly6a*), another STAT1-regulated gene, which persists and remains highly elevated at all subsequent time points (*Figure 2—figure supplement 1C*). Challenge with S.Tm at day 14 triggered an expansion of CM-Ts, but this response was lost by day 60, coinciding with the diminished protection observed at this later timepoint (*Figure 2—figure supplement 1E*). In contrast, SCA-1 expression in NCMs remained upregulated (*Figure 2—figure supplement 1E*).

Notably, the expression of STAT1-mediated TI markers is not solely restricted to BM-derived myeloid cells but also occurs in tissue-resident RPM. RPM loss is evident after 14 days of training (*Figure 2D*), together with increased expression of the STAT1-regulated CXCL9 (*Figure 2—figure supplement 1C,D*). SCA-1⁺ RPM levels fluctuate, elevating at day 14 and increasing again at day 60. Given their pivotal role in curbing the early replication of intracellular pathogens (*Hoffman et al., 2021*), such as *S*.Tm, and their long life span, as opposed to CM and NCM (*Yona et al., 2013*), trained RPM could be an additional determinant in generating long-lasting, localized tissue protection.

From the splenocytes collected during the kinetics experiment, we generated and analyzed bulk RNA-seq data. Principal component analysis (PCA) displayed a substantial shift along PC1 due to BCG training (*Figure 2—figure supplement 1F*). Examination of all differentially expressed genes (DEGs) between BCG and PBS across these time points revealed the greatest transcriptional changes at day 14 and 30 (*Figure 2E*; *Supplementary file 1b*).

To investigate the biological processes represented at these time points, GSEA was performed. At day 14, we detected enrichment for IFNγ response, IFNα response, and IL6-JAK-STAT signaling, with many genes shared across these pathways (*Figure 2F*). This STAT1-dependent program includes several interferon-regulated genes upregulated at day 14, including the *Gbp* gene family, *Ifitm3*, and *Socs3* (*Figure 2G*). GBPs are guanylate-binding proteins involved in antimicrobial activity and inflammasome activation (*Tretina et al., 2019*), while IFITM3 is an interferon-induced transmembrane protein that plays a role in the antiviral response (*Clement et al., 2022*). SOCS3, a suppressor of cytokine signaling, particularly via IL-6/STAT3, is a necessary component for mycobacterial infection control. By blocking IL-6 signaling, SOCS3 allows for TNF and IL-12 signaling to occur, initiating CD4-dependent IFNγ release (*Carow et al., 2013*).

Within the IFNγ response pathway, we noted that associated genes display distinct temporal expression patterns. For example, *Irg1* upregulation is limited to day 14 (*Figure 2—figure supplement*

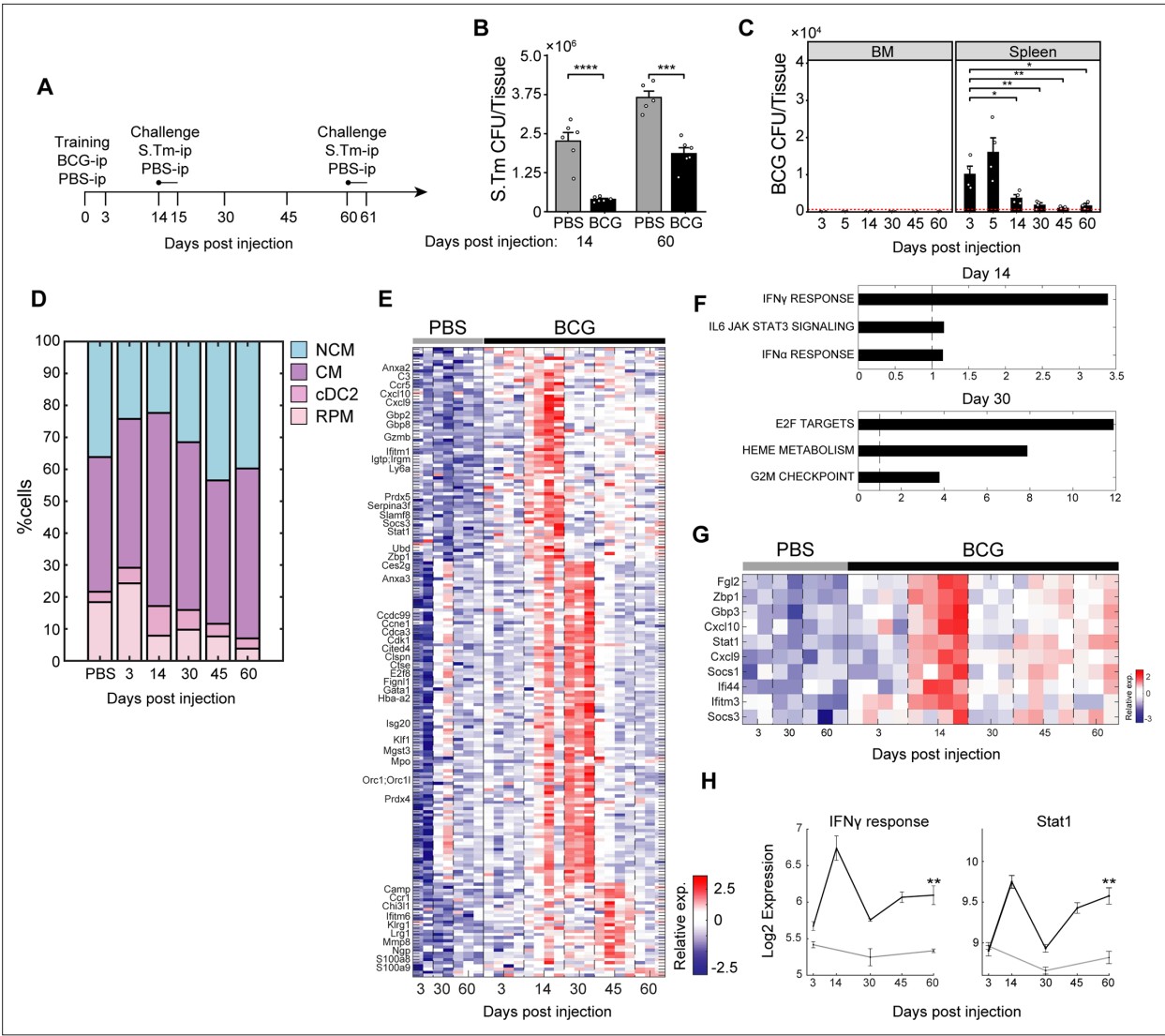

**Figure 2.** Dynamics of TI-associated subsets and signatures indicates early and delayed kinetics. (**A**) Experimental setup tracking TI kinetics over a 2-month interval, including S.Tm challenge at days 14 and 60. (**B**) Splenic *S.Tm* CFU at 24 hr post infection for control and BCG mice at 14- and 60 days after vaccination (n=5–6). (**C**) BCG CFU from spleen and BM of BCG vaccinated mice (n=4) across time points. Red-dotted line indicates limit of detection. (**D**) Contribution of MP populations across time points from control (PBS) and BCG mice (days post injection). Percentage of CM, NCM, and dendritic cells calculated from flow cytometry analysis of CD11b⁺ population. Percentage of RPM calculated from Lin⁻ population (control: n=3, BCG: n=4 in each time point). PBS values are the mean of all time points. (**E**) Heatmap of upregulated genes due to training and relative gene expression ordered according to peak expression time. (**F**) Gene set enrichment analysis of DEGs in days 14 and 30. (**G–H**) Heatmap of IFNγ response genes (**G**) and their average expression dynamics compared to STAT1 expression (**H**). Data in bar and line graphs are presented as mean ± SEM. For bar graph (**B**) and (**C**), each individual point is a biological repeat. For line graph (**H**), significance represents comparison between day 60 control and BCG. Heatmap rows in (**E**) and (**G**) indicate biological replicates. Two-tailed *t*-test used for data in (**B**), (**C**), and (**H**) (*p<0.05, **p<0.01, ***p<0.005, ****p<0.001).

The online version of this article includes the following figure supplement(s) for figure 2:

**Figure supplement 1.** Kinetics gating strategy and marker and gene expression across time and populations.

*1G*) returning to control values for all subsequent time points. This may reflect that IRG1 induction requires bacterial phagocytosis and activation of the TLR-2/MYD88/NFKB axis (*Bomfim et al., 2022*). Conversely, an increase in *Cxcl9* expression is already observed by day 3, plateauing at day 14, and decreasing by day 30, mirroring its induction in CM-Ts in the flow cytometry analysis. Curiously, despite its loss in CM-Ts, the bulk transcriptional expression of *Cxcl9* is still sustained after 2 weeks, although at reduced levels. This may be attributed to its activity in other immune cells or

regulatory mechanisms preventing further translation. Importantly, the IFNγ response mirrored STAT1 activity, showing initial reduction after 2 weeks followed by sustained upregulation for all subsequent time points (*Figure 2H*), reinforcing their coupled role in mediating the effects of TI. Other downstream targets of STAT1, including *Ly6a* (SCA-1), maintain high expression 2 months post vaccination (*Figure 2—figure supplement 1G*).

Distinct from these immune signatures, day 30 was characterized by enrichment of E2F targets, G2M checkpoint, and heme metabolism (*Figure 2E and F*). Interestingly, RPM play a crucial role in iron homeostasis, recycling and storing iron from senescent erythrocytes, preventing accumulation of free iron that can lead to ROS production, oxidative damage, and pathogen utilization (*Vogt et al., 2021*). Given the dynamics of RPM loss by day 14, it is tempting to speculate that their subsequent repopulation is reflected by these processes.

We then investigated whether TI signatures could be maintained through progenitor programming alone. BCG or PBS were injected i.p into mice with either a CD45.1 or CD45.2 background, respectively (*Figure 2—figure supplement 1H*). After 2 weeks, BM was harvested from both, HSCs mixed 1:1, and injected into irradiated mice for BM transplant. Six weeks post transfer, mice were sacrificed and myeloid cells in the spleen were assessed (*Figure 2—figure supplement 1I*). Within each mouse, we found in the spleen a greater fraction of NCMs expressed SCA-1 from trained donors relative to the naive control (*Figure 2—figure supplement 1J*). This persistence of the training signature in a naive host environment demonstrates that SCA-1 expression in NCMs can be maintained through progenitor programming alone, though it does not exclude potential contributions of local tissue signals during normal BCG training. CM-Ts however, were undetectable, consistent with the transient nature of this subset observed in our kinetics data.

## RPM niche is replenished by recruited trained monocytes and by local training of tissue-resident populations

We observed a substantial reduction in RPM numbers upon BCG, followed by the expression of STAT1-mediated TI markers in RPMs. We hypothesized that upon BCG, two possible scenarios for training and replenishment of open niche are possible. The first is self-renewal by the remaining local tissue-resident macrophages who are trained within the tissue, while the other is that BM-derived trained monocytes differentiate and repopulate the open niche. While BM-derived macrophages may adopt signatures and function of their local counterparts, they may also retain aspects of their origin, particularly enhanced inflammatory capacities (*Figure 3A*). To investigate this, we employed MS4A3^Tdtm;CX-3CR1^GFP reporter mice (*Liu et al., 2019*) with a knock-in flox-cre system to selectively label BM-derived monocytes with TdTomato fluorescence. As MS4A3 is distinctly expressed in granulocyte-monocyte progenitors (GMP), only this lineage will be TdTomato positive. These mice were administered BCG or PBS-i.p following the training protocol and sacrificed two weeks later (*Figure 3B*). Effective labeling was determined by flow cytometry, measuring the fraction of MP expressing TdTomato and/or CX3CR1 (*Figure 3—figure supplement 1A*). As expected, the monocyte population were primarily double positive (*Figure 3—figure supplement 1B*). Intriguingly, training increased the percentage of labeled cells across monocytes, suggesting a potential lineage bias towards granulocyte-monocyte progenitors (GMPs). This observation is consistent with prior studies demonstrating that various microbial components can induce short-term differentiation biases in monocytes derived from GMPs or MDPs (monocyte-dendritic progenitors), endowing them with neutrophil- or dendritic cell-like properties (*Yáñez et al., 2017*).

For the majority of RPM, we expected minimal TdTomato expression, representative of self-replenishment during homeostasis. However, already within the control we observed that ~16% of the population were labeled (*Figure 3C*). When this population was depleted during training, an even greater fraction was positively labeled. Accordingly, even under steady-state conditions, BM-derived cells contribute to the resident niche, but when sufficiently diminished post vaccination, active replenishment from the BM does occur.

To determine whether Tdtm+ BM-derived RPM acquired a distinct transcriptional profile due to training compared to the native Tdtm- population, we sorted Tdtm+ RPMs, Tdtm- RPMs, and Tdtm+ CM and NCM, from both trained and naive conditions. These sorted cell populations were then subjected to bulk RNA-seq and subsequent analysis. Crucially, training associated genes were differentially expressed across all sorted cell types (*Figure 3D*; *Figure 3Figure 1C*; *Supplementary file*

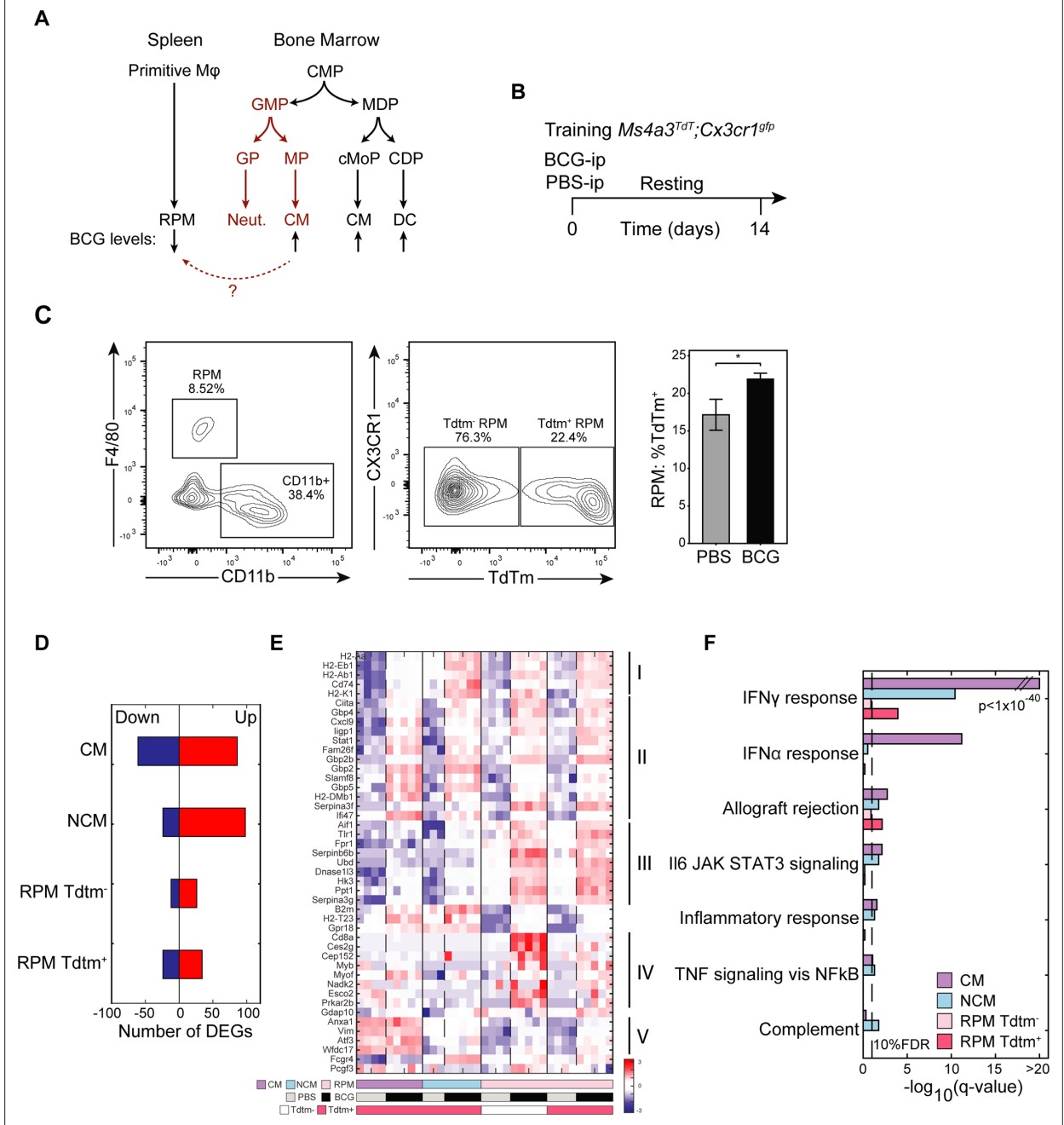

**Figure 3.** RPM niche is replenished by recruited trained monocytes and by local training of tissue-resident populations. (**A**) Scheme representing known myeloid differentiation pathways and potential trans-differentiation of trained CM to RPM. (**B**) Mouse model to track contribution of TI-associated signatures in local and recruited MP populations with lineage tracing. (**C–D**) Flow cytometry analysis of TdTm⁺ or Tdtm⁻ Ly6C⁺ MPs and RPM and quantification of RPM TdTm⁺ subset (n=3). (**D–F**) Number of DEGs of each sorted population (**D**), heatmap of normalized log2 expression from TI-associated DEGs specific to trained RPM populations and gene set enrichment analysis of DEGs in each sorted population (**F**). Data in bar graphs are presented as mean ± SEM. Heatmap rows in (**E**) indicate biological replicates. Two-tailed *t*-test used for data in (**C**) (*p<0.05).

The online version of this article includes the following figure supplement(s) for figure 3:

**Figure supplement 1.** Lineage tracing gating strategy and origin-specific DEGs.

1c). When evaluating specific DEGs upregulated across trained RPM, we identified five gene clusters that varied between CM, NCM, and RPM, as well as genes primarily upregulated in Tdtm⁺ or Tdtm⁻ RPM (*Figure 3E*). Cluster I is composed of MHCII-associated genes (*H2-Aa*, *H2-Eb1*, CD72, etc.), and is enriched in NCM and RPM cells, indicating enhanced antigen presentation that may facilitate the

activation of humoral immunity. Cluster II consists of the STAT-1 regulated TI hallmark genes, including *Cxcl9*, *Gbp2* and *Stat1* and is upregulated in all trained subsets. Genes in Cluster III, such as *Aif1*, *Fpr1*, and *Hk3*, are primarily observed in trained RPM and are involved in response to tissue damage/disruption and immune infiltration. AIF1 is an established marker of macrophage activation and is functionally involved in phagocytosis and membrane ruffling (*De Leon-Oliva et al., 2023*). Similarly, FPR1, a formyl peptide receptor, induces chemotaxis, phagocytic uptake, and reactive oxygen species (ROS) production (*Prevete et al., 2015*). Cluster IV is upregulated in BCG Tdtm⁻ RPM, though the role of most genes detected in this cluster remains unknown in regard to their tissue-specific function and activation. Interestingly, *CD8A* expression, typically relegated to lymphocytes, was observed in this cluster. In monocytes, CD8A can co-engage with FcR, resulting in TNF release (*Gibbings et al., 2007*). Lastly, cluster V, which is primarily activated in CM regardless of training, was also upregulated in RPM due to BCG. Three of the genes within the cluster, *Anxa1*, *Vim*, and *Wfdc17*, have all been shown to dampen excess inflammatory responses through various mechanisms, including suppressing oxidative stress and promoting local resolution (*Sugimoto et al., 2016*; *Håversen et al., 2018*; *Karlstetter et al., 2010*).

GSEA performed across all upregulated genes revealed that the IFNγ response was enriched, followed by IFNα response, allograft rejection, and JAK-STAT signaling (*Figure 3F*). Although the IFNγ signature was most prominent in CM, it was also observed in NCM and in the Tdtm⁺ RPM. This finding suggests that engrafted monocytes differentiating within the niche may maintain a more inflammatory phenotype and a heightened sensitivity to IFNγ activation. However, the BM-derived RPM are not the sole population responsive to training, as the local fraction also upregulates the same genes in clusters I-III. Notably, RPM, as a whole, demonstrate a greater capacity to upregulate the expression of many interferon genes, including *Cxcl9* and *Stat1* (*Figure 3—figure supplement 1D*). Taken together, our results indicate that BCG can reprogram populations and generate training via two separate routes. First, the recruitment of trained progenitors and monocytes within the spleen, which differentiate within a vacant niche, retaining their trained identity. Second, activation directly within the spleen in the context of native tissue-resident RPM, generating tissue-specific protection.

## Transient IFNγ-STAT1 inhibition prevents TI signatures and splenic infection resistance

We observed that STAT1 signaling holds a critical role in training, with its regulated gene expression elevated across the myeloid population. However, BCG is an intact attenuated bacterium that can activate numerous PRRs. To evaluate whether STAT1 is necessary for TI signatures and protection, we vaccinated STAT1-KO mice (*Kernbauer et al., 2012*) with either PBS or BCG-i.p (*Figure 4—figure supplement 1A*). After a two-week period, we assessed the myeloid population and STAT1 regulated genes. We observed a complete absence of expression for both CXCL9 and SCA-1 across myeloid populations (*Figure 4—figure supplement 1B*). This aligns with previous findings that demonstrated compromised acquisition of trained immunity in IFNγR⁻/⁻ mice (*Kaufmann et al., 2018*). Interestingly, RPM, typically depleted following BCG inoculation, remain preserved in STAT1-KO mice (*Figure 4—figure supplement 1C*), suggesting that STAT1-mediated pathways are involved in triggering the cellular death processes that occur during BCG interaction and/or engulfment.

There is however a significant limitation in this mouse model, as STAT1⁻/⁻ leave mice highly susceptible to infection due to a severe compromise of immune homeostasis, limiting our ability to assess TI-mediated protection upon *S.Tm* challenge. In light of our findings that STAT1 signaling is activated shortly after BCG administration, we sought to transiently restrict STAT1 activity at these early time points, also enabling us to investigate the effect of inhibition of STAT1 signaling on training and protection, without affecting STAT1 activation during a secondary *S.Tm* challenge. To accomplish this, we used Fedratinib, a specific inhibitor of JAK2 activation of STAT1 through IFNγ signaling (*Håversen et al., 2018*), and Deucravacitinib, a specific inhibitor of TYK2 activation of STAT1/STAT2 through IFNα/β signaling (*Lé et al., 2022*). First, we administered DMSO (control), Deucravacitinib, and Fedratinib i.p followed by BCG or PBS-i.p 4 hr after. These inhibitors were then injected daily via i.p for the following 4 days, followed by a subsequent 9-day rest period (*Figure 4—figure supplement 1D*). At the 2-week mark, splenocytes were extracted from all mice and analyzed by bulk RNA-seq. Only Fedratinib, not Deucravacitinib, resulted in inhibition of STAT1-mediated TI signatures (*Figure 4—figure supplement 1E*; *Supplementary file 1d*).

We then repeated this experiment, focusing on training phenotypes in the spleen and BM. At the 2-week mark, splenocytes were extracted from all mice for flow cytometry analysis, with CXCL9 utilized as a marker for STAT1-mediated TI signature in CM-Ts. As observed in the transcriptional response, only Fedratinib ablated the CXCL9+ CM-Ts (*Figure 4—figure supplement 1F*). We also sought to determine if these effects were localized solely to the tissue, or if they extended to progenitors in the BM, which expand upon BCG exposure (*Kaufmann et al., 2018*). To ascertain this, we isolated BM from the femur, measuring the percent of LSK+ HSCs (*Figure 4—figure supplement 1G*). Here too, only Fedratinib resulted in suppressing their expansion to levels comparable to the control (*Figure 4—figure supplement 1H*). Conversely, perturbation of IFNα/β signaling with Deucravacitinib lead to no observable changes on trained subsets, suggesting that it is not involved in our BCG-i.p model. Type-I interferon has been established as a training signaling pathway in other contexts, as observed with β-glucan (*Kalafati et al., 2020*), *Candida albicans* (*Huijser et al., 2022*) and LPS (*Zahalka et al., 2022*).

In order to prove that early STAT1 inhibition is sufficient to block the TI protective phenotype, and not just downstream markers, we repeated the inhibitor regime with control or trained mice receiving either Fedratinib or DMSO, with or without *S*.Tm infection after 2 weeks (*Figure 4A*). We then extracted spleens to measure splenic expansion, splenoctye population levels and marker expression with flow cytometry, bulk splenocyte transcription, and *S*.Tm susceptibility. Importantly, while we observed no differences in CFU between control mice with or without Fedratinib, trained mice receiving Fedratinib were significantly more susceptible to infection (*Figure 4B*). Treatment with Fedratinib resulted in diminished recruitment and splenocyte expansion, causing an appreciable reduction in spleen size comparable to the control (*Figure 4—figure supplement 1I*). Accordingly, the balance of monocytes ratios, particularly NCM, was shifted to levels similar to those observed in the DMSO control (*Figure 4C*). In conjunction, the frequency of CXCL9+ CM-Ts was diminished, reflecting a reduction in the subset, while in RPMs, CXCL9 showed decreased expression (*Figure 4D*). Finally, RPM, which typically undergo depletion after training, exhibited significantly enhanced survival, similar to the observations in STAT1-KO mice (*Figure 4C*).

To probe the effects of Fedratinib inhibition beyond myeloid expansion and marker acquisition, we conducted bulk RNA sequencing on total splenocytes isolated from all experimental conditions. PCA of the resulting data revealed two major axes of divergence among the populations: PC1, associated with training, and PC2, associated with the response to *S*.Tm (*Figure 4—figure supplement 1J*). While the PBS-i.p mice treated with either DMSO or Fedratinib were grouped together, the BCG-i.p samples treated with Fedratinib clustered distinctly, shifting closer to the PBS control group. Analysis of downregulated DEGs identified the IFNγ response as the most significantly affected pathway due to JAK2 inhibition by Fedratinib. This included its downstream effector *Stat1*, and other key STAT1-regulated trained immunity genes, such as *Cxcl9/10*, *Irf1*, *Gbp2*, and *Irg1*, across all treated mice (*Figure 4E and F*; *Supplementary file 1e*). To further validate our findings and ensure that the loss of the trained immunity signature was not solely a result of blocking other JAK2-associated pathways, we repeated the inhibitor experiment using recombinant α-IFNγ. Mice were given either α-IFNγ or an isotype control after BCG or PBS vaccination, with injections on days 0, 2, and 4, and assayed 2 weeks post-vaccination. Upon sorting and sequencing CM and RPM from these treated mice, we found that the STAT1 regulated TI signatures were completely ablated with early inhibition (*Figure 4—figure supplement 1K–L*). Although we measured no changes in viable BCG in cultured spleen homogenates from control and treated mice at the 2-week interval (*Figure 4—figure supplement 1M*), early α-IFNγ was sufficient in blocking training signatures. Interestingly, previous studies show that while removal of BCG by long-term antibiotic treatment decreases expansion of progenitor cells, protection against secondary ex vivo challenges remains (*Kaufmann et al., 2018*).

Thus, the impact of Fedratinib and α-IFNγ treatment on the TI phenotype further underscores the pivotal role of IFNγ and the JAK2-STAT1 axis in orchestrating the early programs and signatures required for long-term local and recruited myeloid populations within the tissue. When compromised, remodeling is suppressed and so too protection. Crucially, even though its administration was early and removed after only 5 days administration, inhibition occurred. This indicates that the acquisition of TI in the tissue occurs within a critically narrow temporal window.

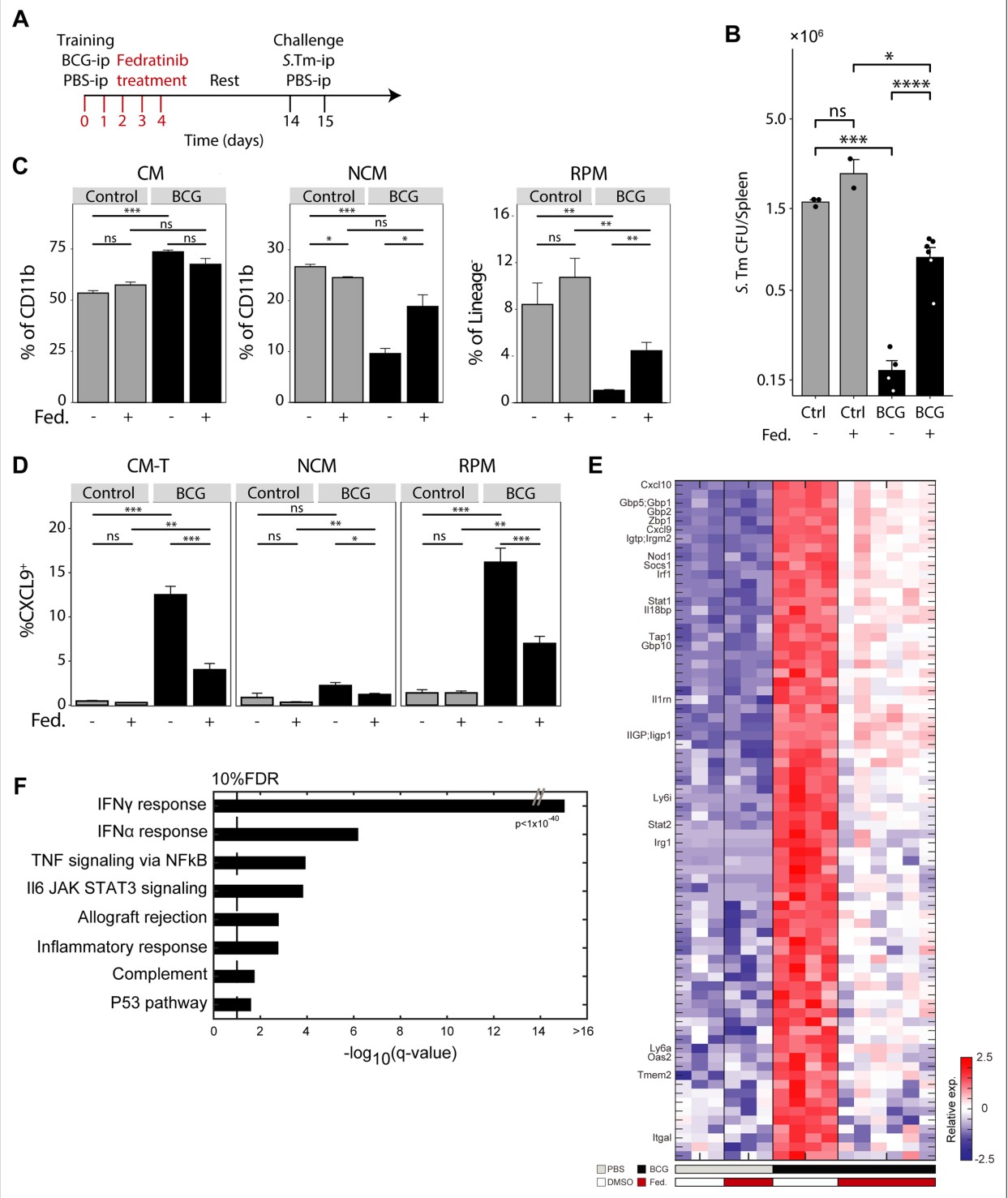

**Figure 4.** Transient IFNγ-STAT1 inhibition prevents TI signatures and splenic infection resistance. (**A**) Mouse model of BCG vaccination with early interferon inhibition using the Fedratinib inhibitor. (**B**) Splenic S.*Tm* CFU for control and BCG mice, with and without Fedratinib inhibitor, 24 hr post infection (n=2–6).(**C**) MP populations from control (gray) and BCG (black) mice, with or without Fedratinib inhibition. Percentage of CM and NCM cells calculated from CD11b⁺ population. Percentage of RPM calculated from Lin⁻ population. (**D**) Percentage of CXCL9⁺ CM-T, NCM, and RPM populations from control (gray) and BCG (black) mice, with or without Fedratinib inhibition (control: n=3, BCG: n=4, control +Fedratinib: n=3, BCG +Fedratinib: n=6). (**E**) Heatmap of normalized log2 expression of DEGs across naive and training conditions. (**F**) Gene set enrichment analysis of DEGs from E. Data in bar graphs are presented as mean ± SEM. For bar graph (**B**) each individual point is a biological repeat. Heatmap rows in E indicate biological replicates. Two-tailed *t*-test used for data in (**B**), (**C**), and (**D**) (*p<0.05, **p<0.01, ***p<0.005, ****p<0.001).

*Figure 4 continued on next page*

*Figure 4 continued*

The online version of this article includes the following figure supplement(s) for figure 4:

**Figure supplement 1.** Training inhibition with Stat-1 KO mice and across interferon inhibition strategies.

## Discussion

TI, or innate immune memory, embodies the capacity of innate immune cells to remember past encounters and modify their subsequent responses, triggered by diverse conditions and stimuli. In-vivo these processes are sustained in progenitor cells within the BM to provide long term TI, with differentiated cells inheriting this memory (*Kaufmann et al., 2018*; *Moorlag et al., 2020*). In this study, we set out to demonstrate how long-term protection events in the BM affords training within the tissue through local interactions with resident populations. We demonstrated how the immune effects of BCG vaccination are differentially imparted across recruited trained monocytes from the BM and diverse MP populations within the spleen, the temporal dynamics governing these processes, and the mechanisms necessary for them to be sustained long-term.

We verified that BCG-i.p inoculation stimulates an expansion of myeloid MP, notably a subset of classical monocytes marked by CXCL9 expression (CM-T) within the spleen. This CM-T population, which peaks at day 14 before declining, exemplifies the broader training response we observed, as our dissection of single-cell training signatures uncovered a fundamental program dominated by IFNγ response and STAT1 regulated genes across diverse populations. Among these, GBP2, CXCL9, and IRG1 represent some of the pathways involved in the protective response: GBP2, a member of the GTPase family, drives macrophages toward an inflammatory state (*Li et al., 2023*) while combating bacterial infection through vacuole disruption and pyroptosis induction (*Meunier et al., 2014*). CXCL9 orchestrates immune cell recruitment and activation at infection sites (*Groom and Luster, 2011*), supporting trained immunity by establishing a pro-inflammatory environment conducive to pathogen clearance (*Joosten et al., 2018*). Meanwhile, IRG1 produces the metabolite itaconate, which serves dual functions in both directly inhibiting pathogen growth and modulating immune responses to prevent excessive inflammation (*Lampropoulou et al., 2016*; *Michelucci et al., 2013*). However, IRG1's role in training appears context-specific, as exposure to training stimuli like β-glucan can actually block its activity (*Domínguez-Andrés et al., 2019*).

By monitoring the kinetics of training over an extended duration, we observed that the initial surge in CM recruitment is confined to the first two weeks of training, potentially signaling inflammatory recruitment triggered by BCG exposure and RPM loss. Concurrently, the presence of CM-Ts starts to diminish at a similar rate and is eventually lost, suggesting that these cells may be differentiated to trained MPs in the tissue.

The depletion of RPMs at 14 days post-vaccination sets in motion a series of complex cellular responses. Notably, we observed an enrichment of heme metabolism at day 30, which may indicate an ongoing process of RPM replenishment. This process typically involves two key mechanisms: the local expansion and repopulation by native populations, and the recruitment of erythroid and myeloid progenitors, including classical and non-classical monocytes, to the organ (*Liao et al., 2018*). These recruited cells, particularly monocytes, can differentiate within the niche, transitioning through a pre-RPM state via heme-mediated pathways (*Haldar et al., 2014*). Interestingly, heme metabolism has also been implicated in the control and pathogenicity of mycobacterial species, particularly *Mycobacterium tuberculosis* (MtB). During systemic infection, MtB activates type I interferon (IFN-I) signaling, which, along with its own virulence factors, disrupts iron transport and uptake (*Khan et al., 2020*).

The presence of TI markers not only in BM-derived myeloid cells such as CM and NCM, but also in RPM, prompted us to uncover how these cells acquired this phenotype. Utilizing Ms4a3TdTm labeled mice, we investigated whether trained RPM were being derived from cells originating in the BM. Unlike CM and NCM, which are predominantly TdTm+, RPM express this lineage tracing marker at steady state around 16%, which was further increased with training. Probing the transcriptional profile of these cells also revealed differences due to origin, with Tdtm+ RPM demonstrating a heightened response to IFNγ stimulation and elevated expression of STAT1. However, both populations exhibit significant upregulation of TI-associated genes post-vaccination. These findings indicate that TI in RPM is not solely due to BM-derived precursors filling a depleted niche but suggests a capacity for both resident and BM-derived RPM to undergo training within the tissue itself. Considering the

importance of these cells in restricting early infection events (*Hoffman et al., 2021*), their capacity for training reveals an additional factor contributing to the TI phenotype. While our examination was limited to a singular tissue-resident macrophage subset, other splenic macrophages like marginal zone macrophages (MZMs) and marginal metallophilic macrophages (MMMs), known to have immunological roles, may similarly exhibit a capacity for being trained. This research can be further broadened to include other resident myeloid populations, to explore how BCG can bestow localized protection independent of central HSC reprogramming.

When investigating the longevity of the trained transcriptional response within the spleen, we noted that Stat-1 and other interferon response genes remained upregulated even 2 months after vaccination. We found one such gene SCA-1 (*Ly6a*), was continuously expressed in NCM due to training, and was still identifiable after a BM transplant. Although SCA-1 is traditionally a marker for upstream progenitors, recent studies link its expression to inflammation, with upregulation noted in B/T cells during *Mycobacterium tuberculosis* (Mtb) infection (*Akter et al., 2022*). Further exploration is warranted to investigate the mechanisms that enable such enduring expression, particularly in this cell type. Also, while *Ly6a* (SCA-1) is unique to mice, a recent study identified a human equivalent termed *Ly6s*, primarily expressed in splenic NCM and regulated by interferon signaling, that was linked to an inflammatory cell phenotype with resistance to viral infections (*Shmerling et al., 2022*).

Prior publications have demonstrated the importance of BCG localization in generating local vs. systemic changes (*Kaufmann et al., 2018*; *Vierboom et al., 2021*), and that BCG's initial presence within the BM is associated with HSC reprogramming and long-term TI. Despite this, while we were able to isolate BCG from splenic tissue, not BM, from all sampled time points, we could still achieve training and protection, including LSK$^+$ subset expansion in the BM. This leads us to hypothesize that the signaling initiated due to BCG can act in trans, affecting system wide changes, and is not solely acquired due to local pathogen interactions in the BM. Supporting this, recent findings suggest that training in alveolar macrophages can also occur through subcutaneous BCG administration, potentially acting via the gut-lung axis (*Jeyanathan et al., 2022*).

Finally, given STAT1's ubiquity among all trained cells, we hypothesized that this transcription factor and its upstream activation via interferon signaling were instrumental in driving BCG-induced trained immunity. We observed that early transient inhibition, specifically targeting IFNγ-mediated STAT1 activation, effectively negated the hallmarks of trained immunity. These include myeloid recruitment, disappearance of RPM, the expression of training markers, LSK$^+$ expansion, transcriptional alterations, and, crucially, heterologous *S*. Tm protection - highlighting STAT1's particular role in orchestrating a protective response against this pathogen. Furthermore, while it is generally known that BCG can reside within the tissue for weeks after inoculation, our findings suggest that even brief STAT1 and IFNγ inhibition is sufficient to disrupt the development of trained immunity, regardless of the pathogen's presence. It is plausible that during the initial stages of BCG exposure and inflammation, IFNγ-secreting cells such as T/NKTs initiate the immunological remodeling needed for training. Should these processes be obstructed at this critical juncture, the opportunity for subsequent training is lost. While our inhibitor protocol was applied for a period of 5 days, the minimal duration required for effective inhibition and comparable outcomes may be even shorter.

In summary, our study emphasizes that examining the tissue, specifically the spleen, as a tissue for probing TI offers valuable insights into the temporal dynamics and signaling cascades that instigate and sustain TI locally, in parallel with established systemic effects. Central to these processes is the IFNγ-STAT1 pathway, which we identified as a key driver in establishing TI, by replenishment of resident naïve MPs with trained recruited and local immune populations. We further delineated that during intraperitoneal vaccination the resultant immune interactions limit BCG dissemination, while still effectively eliciting training. Our findings open new avenues to harness STAT1 pathway induction for optimized training and the importance of vaccines that can induce robust cross-domain protection.

## Methods
### Experimental methods
#### Mice and bacteria strains
C57BL/6 J mice, 7–9 weeks old, were purchased from ENVIGO, housed at the Weizmann Institute pathogen-free facility, and provided with standard food and water ad libitum. The Nr4a1

super-enhancer sub-domain E2-KO (C57BL/6-*Rr39*[em1Ched]/J) mice were purchased from The Jackson Laboratory (#030204; *Thomas et al., 2016*). The mice strains below were kindly provided by the following investigators:

- STAT1-KO mice by Prof. Dr. Mathias Müller (*Kernbauer et al., 2012*).
- Ms4a3Cre[TdTomato]-CX3CR1[GFP] and CD45.1 mice by Prof. Steffen Jung.

All experiments were performed in accordance with the guidelines outlined by the Weizmann Institute Committee on Animal Care (protocol number 00210121–3).

For in vivo training, BCG-Pasteur, generously donated by Dr. Daniel Barkan, was utilized. The *Salmonella enterica* serovar Typhimurium strain SL1344 was used exclusively for all infection challenge experiments.

## Mice training and infection

BCG were grown in Middlebrook 7H9 media (BD) supplemented with Middlebrook OADC (BD) at 37 °C for 1 week to stationary phase. Bacterial aliquots of 1 mL were dispensed to 2 mL cryotubes (Simport) and frozen at –80 C for longterm storage. Prior to inoculation, tubes were thawed, centrifuged (10,000 × $g$, 2 min, RT), with pellet resuspended in 1 mL phosphate-buffered saline (PBS; Sartorius). Bacterial concentration was calculated based on optical density at 600 nm ($OD_{600}$) assuming a concentration of $5 \times 10^8$ CFU/OD, with BCG diluted to $25 \times 10^6$ CFU/mL in PBS. Mice were injected intraperitoneally (i.p) with 200 µl containing $5 \times 10^6$ CFU or PBS (as controls). At given time points, mice were euthanized by $CO_2$, spleens and/or BM from the femur were harvested, and CFU numbers were evaluated by plating serial 10–100-fold dilutions of homogenized spleens or BM suspension on selective 7H9-Middlebrook agar plates.

For the initial challenge as observed in *Figure 1A*, cultures of S.*Tm* were grown in Luria-Bertani (LB) medium (BD) at 37 °C for 16 hr to stationary phase. For all subsequent experiments, S.*Tm* were grown at 37 °C for 16 hr to stationary phase in SPI-2 inducing media (*Stapels et al., 2018*): MgMES media 170 mM 2-(N-morpholino) ethanesulfonic acid (MES) at pH 5.0, 5 mM KCl, 7.5 mM (NH4)2SO4, 0.5 mM K2SO4, 1 mM KH2PO4, 8 mM MgCl2, 38 mM glycerol, and 0.1% casamino acids. Cultures were centrifuged (10,000 × $g$, 2 min, RT), with pellet resuspended in PBS. Bacteria were diluted 10-fold in PBS and concentration calculated based on optical density at 600 nm ($OD_{600}$). Assuming a concentration of $1 \times 10^9$ CFU/OD, S.*Tm* was diluted to $2.5 \times 10^6$ CFU/mL in PBS. Mice were injected intraperitoneally (i.p) with 200 µl containing $1 \times 10^5$ CFU of bacteria or PBS (as controls). Injected bacterial load was verified by CFU. 24 hours post infection, mice were euthanized by $CO_2$, spleens were harvested, and CFU numbers were evaluated by plating serial 10–100-fold dilutions of homogenized spleens on streptomycin LB agar plates.

## BCG CFU

Spleens or BM were homogenized and serially diluted in PBS +0.1% Triton on 7H9 Middlebrook media +OADC with Zeocin and Kanamycin. Plates were incubated for three weeks in a humidified 5% $CO_2$ (*Saeed et al., 2014*) incubator, with CFU determined using an automated colony counter.

## In vivo interferon inhibition

JAK-STAT inhibitors Fedratinib (cat #202893), and Deucravacitinib (cat #555349) (MedKoo Biosciences) were resuspended in DMSO, aliquoted, and stored at –80 C for later use. For injection, a mixture of PEG-300 (Sigma):Tween-80 (Sigma) was prepared at a ratio of 18:1 and filtered using a 0.22 µm filter. For each injection, 10 µl of DMSO with or without the inhibitor was added to 105 µl of the PEG:Tween mix, followed by 180 µl of PBS for a total of 30 µl and injected i.p. Inhibitor concentrations are 1 mg and 0.5 mg per mouse for Fedratinib and Deucravacitinib respectively. Four hours post inhibitor, BCG or PBS-i.p was injected for training. Daily repeat injections of the inhibitors were repeated for four additional days.

For the antibody inhibition experiment, 1 mg of monoclonal αIFNγ antibody (clone: XMG1.2) or isotype control (clone: Rat IgG1) was injected i.p. 4 hr pre BCG vaccination (day 0), then every other day (day 2 and 4).

## Splenocytes and BM isolation and flow cytometry preparation

Spleens and BM from femurs were extracted and stored in cooled FACS Buffer (PBS, 10 mM EDTA, 2% FBS) until further extraction.

For BM extraction, femurs were cut at both ends and placed in 0.5 mL microfuge tube with a small hole in the bottom cut out using an 18 G needle. This tube was then placed in a 1.5 mL microfuge tube and centrifuged (3 min, 500 × $g$, 4 °C). The pellet was resuspended in red blood cell (RBC) lysis buffer for 4 min at room temperature, centrifuged (3 min, 500 × $g$, 4 °C) and re-suspended with FACS buffer. For CFU determination, suspension is used directly for serial dilution. For further flow cytometry processing, FACS buffer containing CD16/CD32 blocking antibodies (BioLegend) is added for a 15 min incubation on ice. All subsequent processing is identical to splenocytes.

The spleens were dissected, mashed against a 70 μm cell strainer (Falcon) and washed with 5 mL of cold FACS buffer. 1 mL of splenocytes were aliquoted to microfuge tubes and centrifuged twice (3 min, 500 × $g$, 4 C). Pellets were re-suspended with RBC lysis buffer (Sigma), incubated for 4 min at room temperature, centrifuged (3 min, 500 × $g$, 4 C) and re-suspended with FACS buffer containing CD16/CD32 blocking antibodies (BioLegend) for 15 min on ice. Cells were centrifuged once more, and pellets were transferred to wells of a 96-well low attachment plate for multi-sample preparation. Subsequently, fluorophore-conjugated antibodies cocktails (listed below) in Brilliant Stain Buffer (BD) were used to resuspend pellets, followed by a 30 min incubation on ice. Cells were washed, re-suspended with 500–1000 μl FACS buffer and passed through a 35 μm cell strainer (Falcon). For absolute quantification of cell populations, 10–50 μl of Precision Count beads (BioLegend) were added to the samples.

## Antibodies used in this study for splenocyte and BM staining

| Epitope | Conjugation | Clone | Company |
|---|---|---|---|
| CD16/CD32 | NA | 93 | BioLegend |
| NK1.1 | APC | PK136 | BioLegend |
| CD19 | APC | 6D5 | BioLegend |
| CD3 | APC | 17A2 | BioLegend |
| Ly6G | APC | 1A8 | BioLegend |
| Ly6C | Alexa Fluor 700 | HK1.4 | BioLegend |
| Ly6C | Brilliant Violet 605 | HK1.4 | BioLegend |
| CD11b | APC/Cy7 | M1/70 | BioLegend |
| CD11c | FITC | N418 | BioLegend |
| CXCL9 | PE | MIG-2F5.5 | BioLegend |
| F4/80 | PE | BM8 | BioLegend |
| F4/80 | Brilliant Violet 421 | BM8 | BioLegend |
| I-A/I-E (MHC-II) | Brilliant Violet 605 | M5/114/15.2 | BioLegend |
| Ly-6A/E (Sca-1) | PE/Cyanine7 | E13-161.7 | BioLegend |
| CD11c | PerCP/Cyanine5.5 | N418 | BioLegend |
| Ter-119 | FITC | TER-119 | BioLegend |
| CD4 | FITC | GK1.5 | BioLegend |
| CD8a | FITC | 53–6.7 | BioLegend |
| Gr-1 | FITC | RB6-8C5 | BioLegend |
| CD45R/B220 | FITC | RA3-6B2 | BioLegend |
| CD11b | FITC | M1/70 | BioLegend |
| CD117/c-Kit | APC | 2B8 | BioLegend |
| Ly-6A/E (Sca-1) | PE-Vio 770 | REA422 | Miltenyi |

## Flow cytometry and sorting for RNA sequencing

Flow cytometry and sorting was performed using the BD FACSAria III (BD). Single cells were sorted into 384-well plates (Eppendorf) containing 2 µl of a solution containing barcoded poly-T primers for reverse transcription (Sigma, Israel) according to the MARS-seq v2.0 protocol (*Keren-Shaul et al., 2019*). For bulk cell capture, 5–10x10$^3$ cells from each population were sorted into tubes containing 300 µl RLT buffer (QIAGEN) with β-mercaptoethanol (BME). Immediately after sorting, plates or tubes were spun down, flash-frozen in a mixture of dry ice and ethanol and stored in –80° C until processing.

## Single-cell RNA-seq library preparation

Single-cell libraries were prepared as described (*Keren-Shaul et al., 2019*). Briefly, mRNA from cells was converted to cDNA alongside barcoding and UMI addition. The cDNA of each plate was pooled followed by second DNA strand synthesis and T7 in vitro transcription. Amplified RNA was fragmented, followed by ligation of partial P5 Illumina sequence, and converted to cDNA. Full sequence of barcoded P5 and P7 of P5 were added by PCR for a sequence ready library. Final libraries were quantified for peak size and concentration using the Agilent TapeStation and Qubit HS DNA Assay kit (Invitrogen), respectively.

## Bulk RNA-seq library preparation

RNA was extracted and cleaned using the RNeasy mini kit (QIAGEN) with DNaseI digestion. Libraries were then prepared according to an in-house MARs-seq or CEL-seq protocol optimized for bulk RNA samples. Final libraries were quantified for peak size and concentration using the TapeStation 4200 (Agilent) and Qubit HS DNA Assay kit (Invitrogen), respectively.

## Library sequencing

Bulk and single cell libraries were diluted to a concentration of 1.8pM and run on the NextSeq platform (Illumina) according to Illumina guidelines, with 75 reads for read1, and 15 reads for read2. A mean of 6 M reads per library for the kinetics data; a mean of 12 M reads per library for IFNγ inhibitor data; and a mean of 3 M reads per library for the MS4a3$^{TdTm}$ bulk sorted population data.

## CD45.1/CD45.2 adoptive transfer

C57BL/6 J mice, expressing CD45.1 or CD45.2, were trained according to our standard protocol using BCG or PBS, respectively. After 2 weeks, BM was isolated from the femur from both mice, resuspended in PBS, and mixed in a 1:1 ratio. Recipient mice (WT C57/BL6J) were irradiated with a single dose of 950 cGy using an XRAD 320 machine (Precision X-Ray [PXI]) and reconstituted the next day via retro-orbital injection of 5x10$^6$ mixed donor BM cells/mouse in 200 µl PBS. Mice were given 6 weeks to allow for reconstitution and repopulation of the hematopoietic system.

## Flow cytometry, CFU, and spleen size analysis and quantification

Flow cytometry data was analyzed using the FlowJo software.

For size quantification, spleens were imaged against a contrasting background, and two-dimensional area was calculated using the ImageJ software.

All graphs quantifying the results from flow cytometry and CFU results were performed using R on RStudio with the Tidyverse package (*Wickham et al., 2019*).

## **Bioinformatics analysis**

### scRNA-seq data analysis

#### Data preprocessing

MARS-seq pipeline (*Keren-Shaul et al., 2019*) was used for demultiplexing, alignment to the genome (mm9), and gene counting by unique molecular identifier (UMI). Overall, we sequenced 1536 cells (768 from control mouse and 768 from trained mouse), with 1474 median UMI count per cell and 668 median genes per cell.

## Data normalization and gene filtration

Only genes with at least one UMI count detected in at least one cell were used. Data was normalized to a library size factor. Factors were calculated by dividing total UMI counts in each cell to the median of the total UMI counts across all cells. Data was transformed to log10 scale (log10(UMI count +1)). Cells with less than 200 UMIs were excluded due to low coverage (24 cells, 9 from naive mouse and 15 from trained mouse). We filtered out cell cycle and ribosomal genes and selected the top 425 most variable genes for further analysis. Variable genes were selected based on fitting of the data to a simple noise model based on the genes mean expression and dispersion (coefficient of variance).

## Data clustering and annotations

Principal component analysis (PCA) was performed on the variable genes, and the first 40 PCs were used for downstream analysis for k-nearest neighbor (KNN)-graph, based on Euclidian distance in PC space. Clustering was performed using Louvain community detection on the KNN-graph (k=20). Overall, we obtained 7 clusters. Cluster identity was inferred using cluster-specific and manually selected genes based on cell classification literature.

## Bulk RNA-seq data processing and normalization

MARS-seq pipeline was used for samples demultiplexing, alignment to the genome (mm9), and gene counting. Data was normalized to a library size factor. Factors were calculated by dividing the total number of reads from each sample to the median total number of reads across all samples. These procedures were done for each dataset alone.

## Kinetics data

Data was transformed to log2 scale, and minimal expression threshold was set to 3. Replicate samples of each condition were averaged, except for 1 sample that was excluded due to low coverage (<100 k reads; BCG 30d replicate 3). Preceding PCA analysis genes were centered and normalized to a mean of 0 and a standard deviation of 1. To identify genes that were up-regulated due to training, we calculated the differences between the integrals of each gene in BCG relative to PBS along time. The differences across all genes were approximately normally distributed, with a mean of 0.2 and a standard deviation of 10.11. Genes with more than 3 standard deviations above the mean were defined as up-regulated due to training.

## Bulk inhibitor data

Data was transformed to log2 scale, and minimal expression threshold was set to 3. Two Fedratinib samples were excluded from analysis due to technical issues during injection that resulted in a lack of inhibition. Heatmap was generated using DEGs calculated by ANOVA (5% FDR and a minimal twofold; 167 genes).

## Fedratinib inhibitor data

Data was transformed to log2 scale, and minimal expression threshold was set to 4. One sample was excluded due to low coverage (<100 k reads; PBS +S.*Tm*+Fedratinib replicate 3). Preceding PCA analysis genes were centered and normalized to a mean of 0 and a standard deviation of 1. PCA analysis was performed on DEGs calculated using two-sided t-tests between all relevant conditions: control vs. trained samples, trained with or without inhibitor, control with or without inhibitor, uninfected vs. infected, infected with or without training, infected vs. infected with training with inhibitor, and infected with training vs. infected with training with inhibitor (5% FDR; 453 genes). Heatmap was generated using two-sample t-test between control and BCG-trained samples (1% FDR).

## αIFNγ inhibitor data

Data was transformed to log2 scale, and minimal expression threshold was set to 3. Two sample was excluded due to low coverage (<250 k reads CM BCG +Isotype replicate 1 and RPM BCG +Isotype replicate 4). Heatmap genes were selected from cluster I and II from the lineage tracing experiment representing shared interferon/STAT-1 upregulated genes.

### MS4A3^Tdtm bulk population data

Data was transformed to log2 scale, and minimal expression threshold was set to 3. One sample was excluded due to low coverage (<150 k reads; control NCM-Tdtm+ replicate 4). Preceding PCA analysis genes were centered and normalized to a mean of 0 and a standard deviation of 1. DEGs between control and trained mice were calculated using two-sided t-test between all control samples versus all trained samples (from all sorted populations together; 10% FDR, 382 genes).

## Acknowledgements

This research was supported by the Israel Science Foundation (ISF grant No. 1289/22), the Minerva Foundation with funding from the Federal Ministry for Education and Research, the Dr. Barry Sherman Institute for Medicinal Chemistry, and the Shimon and Golde Picker Weizmann Annual Grant.

## Additional information

### Funding

| Funder | Grant reference number | Author |
|---|---|---|
| Israel Science Foundation | 1289/22 | Roi Avraham |
| Minerva Foundation | | Roi Avraham |

The funders had no role in study design, data collection and interpretation, or the decision to submit the work for publication.

### Author contributions

Aryeh Solomon, Conceptualization, Data curation, Formal analysis, Investigation, Methodology, Writing – original draft, Writing – review and editing; Noa Bossel Ben-Moshe, Data curation, Investigation; Dotan Hoffman, Data curation, Investigation, Methodology; Sébastien Trzebanski, Leia Vainman, Investigation, Methodology; Dror Yehezkel, Data curation; Mihai G Netea, Data curation, Methodology; Roi Avraham, Conceptualization, Supervision, Funding acquisition, Writing – original draft, Project administration, Writing – review and editing

### Author ORCIDs

Aryeh Solomon ⓘ https://orcid.org/0000-0001-7647-5465
Sébastien Trzebanski ⓘ https://orcid.org/0000-0002-0647-6814
Mihai G Netea ⓘ https://orcid.org/0000-0003-2421-6052
Roi Avraham ⓘ https://orcid.org/0000-0002-9098-3885

### Ethics

All of the animals were handled according to approved institutional animal care and use committee (IACUC) protocols os the Weizmann Institute of Science (protocol number 00210121-3).

Reviewer #1 (Public review): https://doi.org/10.7554/eLife.100922.3.sa1
Reviewer #2 (Public review): https://doi.org/10.7554/eLife.100922.3.sa2
Author response https://doi.org/10.7554/eLife.100922.3.sa3

## Additional files

### Supplementary files

Supplementary file 1. Supplementary tables indicating significant DEGs. (**a**) Training induced DEGs from single-cell experiment. For each population (CM, NCM, mcDC, and NK cells), significantly differentially expressed genes (10% FDR) between naive and trained cells are presented, with log-ratio values, and corresponding p-value and q-value.(**b**) Training induced DEGs from the kinetics experiment. Total splenocyte log2-ratio expression from both naïve (PBS) and trained (BCG) vaccinated mice at days 3, 14, 30, 45, and 40 days after vaccination. Replicates are marked at the

end of the sample name.(**c**) Significantly expressed DEGs (10% FDR) between naïve and trained splenic CM, NCM, and Tdtm$^{+/-}$ RPM. Log-ratio values, and corresponding p-value and q-values presented.(**d**) Training induced significant DEGs (10% FDR) in total splenocytes, between naïve and trained mice, with and without Type I and II interferon inhibition. Values are log2-ratio expression per replicate. Replicates are marked at the end of the sample name.(**e**) Training induced significant DEGs (10% FDR) in total splenocytes, between naive and trained mice, with and without Fedratinib inhibition. Values are log2-ratio expression per replicate. Replicates are marked at the end of the sample name.

MDAR checklist

## Data availability

All RNA-seq data have been deposited in NCBI's Gene Expression Omnibus (GEO) under the super-series accession number GSE252014.

The following dataset was generated:

| Author(s) | Year | Dataset title | Dataset URL | Database and Identifier |
|---|---|---|---|---|
| Aryeh S, NoaBossel BM, Dotan H, Dror Y, Leia V, Mihai N, Roi A, Sebastien T | 2025 | Early and Delayed STAT1-Dependent Responses Drive Local Trained Immunity of Macrophages in the Spleen | https://www.ncbi.nlm.nih.gov/geo/query/acc.cgi?acc=GSE252014 | NCBI Gene Expression Omnibus, GSE252014 |

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
