## [Editor Report · eLife Assessment]

This **important** work advances our understanding of the contribution of tissue-resident immune cells to trained immunity phenotypes. The evidence supporting the claims is **convincing**, with results that will be of interest to immunologists and scientists studying the host-pathogen interface.

---

## [Referee Report · Reviewer #1 (Public review)]

Summary:

In the submitted manuscript, Solomon et al carefully detail shifts in tissue-specific myeloid populations associated with trained immunity using intraperitoneal BCG injection as a model for induction. They define the kinetics of shifts in myeloid populations within the spleen and the transcriptional response associated with IP BCG exposure. In lineage tracing experiments, they demonstrate that tissue-resident macrophages, red-pulp macrophages (RPM) that are rapidly depleted after BCG exposure, are replenished from recruited monocytes and expansion of tissue-resident cells; they use transcriptional profiling to characterize those cells. In contrast to previous descriptions of BCG-driven immune training, they do not find BCG in the bone marrow in their model, suggesting that there is not direct training of myeloid precursor populations in the bone marrow. They then link the observed trained immunity phenotype (restriction of heterologous infection with ST) with early activation of STAT1 through IFN-γ.

Strengths:

The work includes careful detaining of shifts and origins of myeloid populations within tissue associated with trained immunity and is a meaningful advance for the field.

Caveats:

Given that the authors demonstrate that BCG persists in the spleen, it is possible that some level of BCG persistence in the spleen is a necessary contributor (together with signaling through STAT1) to the observed tissue-specific T1 phenotype.

Whether ongoing signaling through the axes are required for ongoing protection is not specifically addressed in this work. There is recent work by other groups that partially addresses these caveats, and it would be helpful context to reference those papers.

---

## [Referee Report · Reviewer #2 (Public review)]

Summary:

In this study, Solomon and colleagues demonstrate that trained immunity induced by BCG can reprogram myeloid cells within localised tissue, which can sustain prolonged protective effects. The authors further demonstrate an activation of STAT1-dependent pathways.

Strengths:

The main strength of this paper is the in-depth investigation of cell populations affected by BCG training, and how their transcriptome changes at different time points post-training. Through use of flow cytometry and sequencing methods, the authors identify a new cell population derived from classical monocytes. They also show that long-term trained immunity protection in the spleen is dependent on resident cells. Through sequencing, drug and recombinant inhibition of IFNg pathways, the authors reveal STAT1-dependent responses are required for changes in the myeloid population upon training, and recruitment of trained monocytes.

Weaknesses:

A significant amount of work has already been performed for this study. No significant weaknesses were found.

Comments on revisions:

I thank the authors for carefully considering all reviewer comments. I have no further recommendations for the authors.

---

## [Author Response]

The following is the authors’ response to the original reviews.

**Reviewer #1:**
(1) “…Given that the focus in the paper is on tissue-specific immune training, it would be helpful to know whether the ongoing presence of BCG at low levels in the profiled tissue contributes to the trained immunity phenotypes observed.”….“To address point 1, the authors could treat with anti-BCG antibiotics at 2 or 4 weeks post-BCG exposure and profile the impact on trained immunity phenotypes.”

We thank the reviewer for this important comment. The experiment suggested by the reviewer is to treat with abx to remove BCG from the tissue from the first week post challenge for the duration of four weeks. In previous work, Kaufmann et al (PMID: 29328912) showed that after a month of antibiotics, BCG levels are reduced, but residual BCG levels still remains. Accroding to their results, while antibiotic treatment reduces the training phenotype of LKS^+^ HSC expansion in the bone marrow, protection against TB was maintained during ex-vivo challenge of BMDMs.

In our experiments, we are concerned that antibiotic treatment will only change the dynamics of BCG clearance, but residual BCG will remain and will limit our interpretation. Furthermore, examining the transcriptional changes we observed at early timeponts after BCG may not be relavant at 1 month post antibiotics.

As an alternative approach, we refer to our results with an antibody to block early IFNg signaling (1-5 days; Figure S4 K-M). Here, although BCG levels are comparable between treatment and control groups, we were unable to detect any TI-related transcriptional signatures upon early aIFNg treatment. This indicates that that residual BCG is not sufficient for the TI phenotype in the spleen. We now emphasize this point in the revised version of the manuscript (see lines 335-339).

(2) “Related to the point about BCG above, it would be helpful to understand whether this is a specifically time-limited requirement when trained immunity is first induced, or whether ongoing signaling through this axis is required for maintenance of the observed trained immunity phenotypes.”… “To address point 2, authors could treat with the inhibitor at 2 weeks and/or 4 weeks post-BCG and profiling later transcriptional and/or *Salmonella* growth phenotypes.”

We thank the reviewer for his comment, but respectfully claim that this experiment might not be feasible. As IFNg signaling is directly required for control of Salmonella infection, we are concerned that late IFNg inhibition will also directly affect the response to *Salmonella* challenge and control. Thus, in our experiments, to ensure that treatment only affects the response to BCG challenge, we were careful to limit aIFNg treatment to the early time points and allowed long resting period before *Salmonella* challenge.

Furthermore, inhibition of IFNg at late time point was already tested in both Lee et al, and Tran et al. (PMID: 38036767, 38302603). The authors show that late blockage of IFNg signalling (days 14-21) is sufficient to prevent protection during a viral challenge. This would indeed imply that ongoing signalling is necessary in this context to generate protection, specifically also late signalling events. Furthermore, Lee at al., also observed a biphasic activation pattern of cytokines and recruited cells, suggesting that rather than continuous activation, sequential cell activation and signalling may be occurring.

Respectfully, in our experiments we focus on the early time points based on our observations of early recruitment of CM-T cells (Figure S2. C-D). This was our main findings of this paper. We agree with the reviewer that future experiments are required to compare the differences in cell populations that are invovled in the early vs. late trained phenotpe dynamics.

Minor points:Experimental conditions for the shown data are not consistently clear from the figure legends- would add more detail about the biological conditions.

OK – done

Figure 3E missing units on the legend

OK – done

Figure 4C middle panel missing y-axis label

OK – done

Line 40- remove "both"

OK- done

Line 156- Language could be clearer about what was described previously in contrast to the results shown in this work

We have modified the text accordingly in the revised manuscript

**Reviewer #2:**
“A significant amount of work has already been performed for this study. The work is rich with data and description.”

We thank the reviewer for acknowledging the importance of our work.

Minor comments for the authors to consider:“BCG is widely recognised to induce trained immunity. In this study, *Salmonella* is used as secondary infection event. Why? What is role of *Salmonella* in this study? Does this study contribute to our understanding of the *Salmonella* infection process? What does this tell us about *Salmonella*/vaccines? Is there any evidence that BCG protects against *Salmonella* infection? “

We thank the reviewer for this important comment. We now added to the introduction and the discussion the relevance of our study to the potential of BCG and trained immunity as an alternative heterologous vaccine approach to traditional vaccines that require strain-specific vaccine for each pathogen (lines 49-55 of the revised manuscript).

“Figure 1E. RPM cannot be detected by scRNAseq?”

The reviewer is correct. we excluded RPMs from the scRNA-seq analysis. As we discuss in the manuscript (lines 94-96), and in our previous publication (PMID: 34788598), RPM activation involves rapid cell death. As we are analyzing by scRNA-seq two weeks after BCG challenge, we only measured scRNA-seq of CD11b+ cells, which exclude RPMs, as we were worried that our transcriptional data would represent transcriptional signatures of dying cells, making interpretation of the data difficult.

“Figures 1H and I. The CM-T macrophages are not represented? Are they contemplated within the CM population? Would be useful to see the contribution of CM-T to the total CM DEGs/pathways.”

The reviewer is correct. CM-T cells are evident only after BCG challenge. Because of this, our analysis of DEGs induced in monocytes by BCG requires analysis of all monocytes together. Thus, we were careful throughout the manuscript to refer to CM when analyzing bulk RNA-seq data.

“Lines 104-117. Can the authors summarise or move the text in this paragraph to discussion? Although it provides important context, it cuts the line of thought and reduces comprehension of this section. “

OK – we moved this section to the discussion in the revised manuscript.

“Line 127. Is it Fig 1I or 1F that the authors are referring to? “

The reviewer is correct, and we changed the text in the revised manuscipt accordingly.

“Figure 1J. x-axis labels CM cells but both text and figure legend refer to this panel as CM-T. If this is the case, please show data for CM and CM-T separately.”

Please see our earlier point above that limits these analyses. As such we have also edited the text and figure legend to reflect this.

“Lines 136-139. Please indicate that this can be found in Fig 1J.”

OK – indicated in the revised manuscript

“Line 152. Please add that STm infection occurred at 14 and 60 days post training.”

OK – added

“Lines 162-163. This is repeated from lines 89-90, maybe the reduction of RPMs can be only highlighted in this section so that the previous section can be just focused on the new CM-T population?”

The reviewer is correct - we removed the mention of RPMs here, and mention them only later in the revised manuscript.

“Line 163. The recruitment is CM or CM-T cells? Since they express CXCL9 (line 165 and Fig1J) could this be used as a marker for the CM-T population at this time point?”

The reviewer is correct, and we thank him for this important comment. We now indicate that CXCL9+ is a marker for the CM-Ts population here and throughout the revised manuscript (lines 153-155 of the revised manuscript).

“Line 173. The loss of CXCL9 at 60 dpi means that CM-T population disappears/reduces or returns to CM only? If the population is reduced, could it be related to the reduced STm infection control at 60 days?”

OK– done. Referred to these cells as CM-Ts and suggested a correlation with protection loss in the text (lines 160-162 of the revised manuscript).

“Figure 2D. Can the authors show if there is variation in the myeloid populations after PBS injection at different time points? Are the percentages shown only at 3 dpi? It is curious that at 30 dpi the transcriptome has a significant change for certain genes.”

There are indeed variations across the PBS time points samples, which we demonstrate in Figure S2B. The percentages shown in the main figure for PBS reflect the mean of all time points, this is now stated in greater clarity in the revised manuscript (lines 151-152). We also noted an increase in the cell cycling genes at D30 for the control mice as well, and while still significant in BCG, we limited interpretation accordingly.

“Line 208. The authors can highlight that the expression of STAT1 follows the same pattern as IFNg. Maybe even present the graphs side by side?”

The reviewer is correct, and we have implemented their suggestion as such in the updated text (lines 192-195) and figure (Fig. 2H).

“Line 213. Authors mention a replenishment of the RPM population - what time point are you referring to? At 60 dpi the population seems to be halved compared to 14 dpi. Later (line 230), authors refer to the replenishment as a repopulation by other cell types - is repopulation more correct than replenishment?”

The reviewer is correct, and we thank the reviewer for this important comment. We now changed replenishment to repopulation (lines 95, 201), which is more accurate given the continued decreased percentage at later time points.

Lines 214-222. It is not clear what is the conclusion from these experiments: is the recruitment of progenitors from the BM or by local signals?

The reviewer is correct, we agree that the wording in the initial manuscript was imprecise. This experiment specifically tests whether trained bone marrow progenitors can sustain the observed TI signatures in a naive environment. By transplanting trained bone marrow into naive hosts, we demonstrate that progenitor programming alone is sufficient to maintain long-term SCA-1 expression in NCMs, without requiring ongoing local tissue signals. We now better clarify this text in the revised manuscript (lines 202-212).

“Line 333-334. Where is the data that shows that upon Fedratinib RPMs have enhanced survival?”

OK – We now indicate the figure in the revised manuscript.